# Arbovirus vectors of epidemiological concern in the Americas: A scoping review of entomological studies on Zika, dengue and chikungunya virus vectors

Reilly Jones[1], Manisha A. Kulkarni[2], Thomas M. V. Davidson[3], RADAM-LAC Research Team[1,2,4,5,6,7,8,9]¶, Benoit Talbot[2]*

1 Dalla Lana School of Public Health, University of Toronto, Toronto, ON, Canada, 2 School of Epidemiology and Public Health, University of Ottawa, Ottawa, ON, Canada, 3 Department of Human Biology, University of Toronto, Toronto, ON, Canada, 4 Toronto General Hospital Research Institute, University Health Network, Toronto, ON, Canada, 5 Center for Investigation in Tropical Microbiology and Parasitology, Universidad de los Andes, Bogota, Colombia, 6 Department of Mathematics and Statistics, York University, Toronto, ON, Canada, 7 Group for Investigation in Applied Genetics (GIGA), IBS, UNaM-CONICET, Posadas, Argentina, 8 School of Medicine, Universidad Laica Elroy Alfaro de Manabí, Manta, Ecuador, 9 Center for Investigation of Arthropod Vectors, Instituto Nacional de Investigación en Salud Pública, Quito, Ecuador

¶ Membership of the RADAM-LAC Research Team is listed in the Acknowledgments
* benoit.talbot@uottawa.ca

**Data Availability Statement:** All relevant data are within the paper and its Supporting Information files

## Abstract

### Background

Three arthropod-borne viruses (arboviruses) causing human disease have been the focus of a large number of studies in the Americas since 2013 due to their global spread and epidemiological impacts: Zika, dengue, and chikungunya viruses. A large proportion of infections by these viruses are asymptomatic. However, all three viruses are associated with moderate to severe health consequences in a small proportion of cases. Two mosquito species, *Aedes aegypti* and *Aedes albopictus*, are among the world's most prominent arboviral vectors, and are known vectors for all three viruses in the Americas.

### Objectives

This review summarizes the state of the entomological literature surrounding the mosquito vectors of Zika, dengue and chikungunya viruses and factors affecting virus transmission. The rationale of the review was to identify and characterize entomological studies that have been conducted in the Americas since the introduction of chikungunya virus in 2013, encompassing a period of arbovirus co-circulation, and guide future research based on identified knowledge gaps.

### Methods

The preliminary search for this review was conducted on PubMed (National Library of Health, Bethesda, MD, United States). The search included the terms 'zika' OR 'dengue'

**Funding:** This study was funded by a grant from the Canadian Institutes for Health Research (CIHR) and International Development Research Centre (IDRC)'s CIHR-IDRC Canada-Latin America and Caribbean Zika Virus Research Program to the RADAM-LAC Research Team, and an Early Researcher Award from the Ontario Ministry of Research, Innovation and Science to MK.

**Competing interests:** The authors have declared that no competing interests exist.

OR 'chikungunya' AND 'vector' OR 'Aedes aegypti' OR 'Aedes albopictus'. The search was conducted on March 1st of 2018, and included all studies since January 1st of 2013.

## Results

A total of 96 studies were included in the scoping review after initial screening and subsequent exclusion of out-of-scope studies, secondary data publications, and studies unavailable in English language.

### Key findings

We observed a steady increase in number of publications, from 2013 to 2018, with half of all studies published from January 2017 to March 2018. Interestingly, information on Zika virus vector species composition was abundant, but sparse on Zika virus transmission dynamics. Few studies examined natural infection rates of Zika virus, vertical transmission, or co-infection with other viruses. This is in contrast to the wealth of research available on natural infection and co-infection for dengue and chikungunya viruses, although vertical transmission research was sparse for all three viruses.

## Introduction

Arboviruses, or arthropod-borne viruses, comprise a diverse group of viruses mostly transmitted by mosquitoes and ticks, including globally spreading viruses causing human disease, such as Zika, dengue, and chikungunya viruses. The term arbovirus does not encompass a taxonomically distinct group, but these viruses have similar life-history and transmission patterns that make information gleaned from one virus potentially useful to the understanding, and therefore prevention and control, of the others.

Since its identification in Uganda in 1947, Zika virus (*Flavivirus*, *Flaviviridae*) has been, until recently, confined only to Africa and Asia [1]. The virus ultimately reached the Americas in late 2014, resulting in the declaration of a Public Health Emergency of International Concern by the World Health Organization [2]. To date, 86 countries have reported evidence of mosquito-transmitted Zika virus infection. [3] Brazil currently faces the greatest burden of Zika virus infections [4]. Dengue fever, caused by four different serotypes of dengue virus (*Flavivirus*, *Flaviviridae*) is the most common arboviral disease that affects humans– 50 million people contract it each year, and an estimated 22,000 die from severe dengue [5]. Dengue is hyperendemic in the Americas, with cyclic epidemics occurring every three to five years [6]. Chikungunya virus (*Alphavirus*, *Togoviridae*) was first isolated in Tanzania in 1952 [7]. In the early 2000s, chikungunya virus cases and outbreaks were identified in countries in Africa, Asia, and Europe [7]. In 2013, it emerged in the Americas in Saint-Martin, and within the first year, over a million new cases were reported, spreading to 45 countries in the Latin American and Caribbean region [8].

A large proportion of Zika, dengue, and chikungunya viral infections are asymptomatic [9–11]. However, all three viruses are associated with moderate to severe health consequences in a small proportion of cases, with neonates, young children and/or older age groups at higher risk. Symptoms of Zika viral infection include rash, fever, arthralgia, and conjunctivitis [11]. More importantly, since its initial emergence in the Americas, Zika virus has been confirmed as a cause of congenital abnormalities (in infants born to women infected with Zika virus

during pregnancy) and as a trigger of Guillain-Barré Syndrome [12]. Symptoms of dengue viral infection include rash, fever, arthralgia, and nausea. Some of the more severe symptoms of dengue viral infection may include deadly hemorrhage and plasma leak [9]. Symptoms of chikungunya viral infection include rash, fever, and arthralgia that may persist for an extended duration [7].

Two mosquito species, *Aedes aegypti* and *Aedes albopictus*, are among the world's most prominent arboviral vectors. *Ae. aegypti* originated in sub-Saharan Africa as a sylvatic species and was introduced to the Americas via ships soon after European arrival in the 1400s [13]. The species became domesticated and is now endemic to the Americas and the Asia-Pacific. The range of *Ae. albopictus* was restricted to Asia until the latter part of the 20th century. It is thought to have been introduced to the Western hemisphere through a shipment of used tires in 1985 and has expanded its territory to over 40% of the world's landmass over the course of the past 30 years [14–16].

This review summarizes the state of the literature surrounding the vectors of Zika, dengue and chikungunya viruses and factors affecting virus transmission in the Americas, with a focus on public health implications. Waddell et al. [17] conducted a comprehensive scoping review of the Zika virus literature in 2016. However, the authors identified a limited scope of literature on vector studies, and none specifically looked at vector populations of the Americas, highlighting the need for a scoping review focusing on this area given its relevance in understanding arboviral disease risk in the region. This scoping review aims to identify and characterize the literature pertaining to mosquito species vector competence and aspects of virus transmission dynamics in the Americas since the introduction of chikungunya in 2013. This timeframe includes the introduction of Zika virus and the ongoing co-circulation of three globally spreading arboviruses, namely Zika, dengue and chikungunya viruses.

## Methods

This study's search strategy and data extraction protocol were developed *a priori*. The list of definitions for each search term and the data characterization and utility form are available upon request. The review was conducted using PRISMA guidelines for scoping reviews [18]. See S2 Table for this scoping review's checklist. The preliminary search for this review was conducted on PubMed (National Library of Health, Bethesda, MD, United States). The search included the terms 'zika' OR 'dengue' OR 'chikungunya' AND 'vector' OR 'Aedes aegypti' OR 'Aedes albopictus'. The search was conducted on March 1st of 2018, and included all studies since January 1st of 2013. We chose the year 2013 as a start date for our search to reflect the timing of chikungunya virus spread to the Americas, followed in 2014 by Zika virus. These years are thus characterized by co-circulation of multiple globally spreading arboviruses in the region. Upon selection of potentially relevant articles, studies were characterized according to main characteristics including study setting, virus of interest, study design, methods of mosquito collection and analysis, vector species discussed, and main findings. Zotero (Center for History and New Media, George Mason University, United States) was initially used for title and abstract screening. All studies were subsequently transferred to Excel (Microsoft Corporation, Redmond, WA, United States) for data characterization and extraction. Two independent reviewers completed each step of the review following the broad initial screening, which was conducted by one reviewer.

Articles were selected if they were related to vector species composition and/or virus transmission dynamics, if they were related to Zika, dengue and/or chikungunya arboviruses, and if they were related to the ongoing virus circulation in the Americas. Other inclusion criteria included availability of an English language version and investigation of primary data. Studies

that specifically examined the impacts of vector control measures, or studies that were unrelated to vector-borne aspects of disease, vector competence or entomological measures, were excluded due to the degree of scope expansion that would be caused by their inclusion.

## Results

### Descriptive statistics of scoping review

The search yielded 6267 results. All records were screened, and 5919 were not deemed relevant based on title and abstract content. A total of 348 screened full-text studies were examined for eligibility, and ultimately 96 studies were included in the scoping review (Fig 1; S1 Table). The vast majority of studies were performed exclusively in the field, in the laboratory, or using a modelling framework, and most studies were conducted exclusively on *Ae. aegypti* (Table 1). Studies focusing exclusively on dengue virus were the most numerous, followed by studies focusing exclusively on Zika virus, while studies focusing on chikungunya virus or on a combination of arboviruses were the least numerous (Table 1). Studies on virus transmission dynamics were the most numerous, while studies on aspects of both vector species composition and virus transmission dynamics were the least numerous (Table 1). The average monthly number of studies hovered between 0 and 2 from 2013 to 2016, then increased to 3 or more in 2017 and 2018 (Fig 2), closely reflecting the introductions of chikungunya and Zika viruses in the Americas and subsequent epidemics, respectively.

### Vector species composition

**Zika virus.** There is extensive evidence that *Ae. aegypti* mosquitoes are able to transmit Zika virus in both the laboratory [19–29] and in the field [30–32]. *Ae. albopictus* mosquitoes were also able to transmit Zika virus in experimental studies [22,23], but studies in which both *Ae. aegypti* and *Ae. albopictus* were captured found no Zika virus-infected *Ae. albopictus* [31,32]. Gendernalik et al. [33] and O'Donnell et al. [25] report that *Ae. vexans* mosquitoes are also experimentally competent vectors of Zika virus, but no studies indicated natural *Ae. vexans* infection with Zika virus. *Cx. quinquefasciatus* has been identified by predictive models as a potential vector for Zika virus [34], as have *Sabethes* and *Haemagogus spp.* [35]. Seven studies found that *Cx. quinquefasciatus* mosquitoes were refractory to Zika virus when exposed to infectious blood meals [29,36–42]. Ferreira-de-Brito et al. [31] reported that no *Cx. quinquefasciatus* captured in Brazil were positive for Zika virus. In contrast, Guedes et al. [43] detected Zika virus in the midgut, salivary glands and saliva of artificially fed *Cx. quinquefasciatus* captured in Brazil, using RT-PCR and transmission electron microscopy. The same study also reported Zika virus isolated from two field-caught *Cx. quinquefasciatus* in Brazil.

**Dengue virus.** *Ae. albopictus* [44–47] and *Ae. aegypti* [27,45,46,48–50] are both experimentally competent to transmit dengue virus. Infection by the virus is observed in field populations of *Ae. albopictus* [51–54], *Ae. aegypti* [51,52,54–62] and *Cx. quinquefasciatus* [56], although the latter was not identified as a competent vector species experimentally.

**Chikungunya virus.** *Ae. aegypti* [46,63–68], *Ae. albopictus* [46,64,66–69], *Aedes terrens* [70], and *Haemagogus leucocelaenus* [70] are all experimentally competent to transmit chikungunya virus. Chikungunya virus transmission in *Ae. aegypti* has also been observed in the field [30,59,71,72].

### Virus transmission dynamics

**Vector competence factors.** Four studies measured the effect of temperature on vector competence [47,63,64,73]. Adelman et al. [63] found that under silenced RNAi conditions, *Ae.*

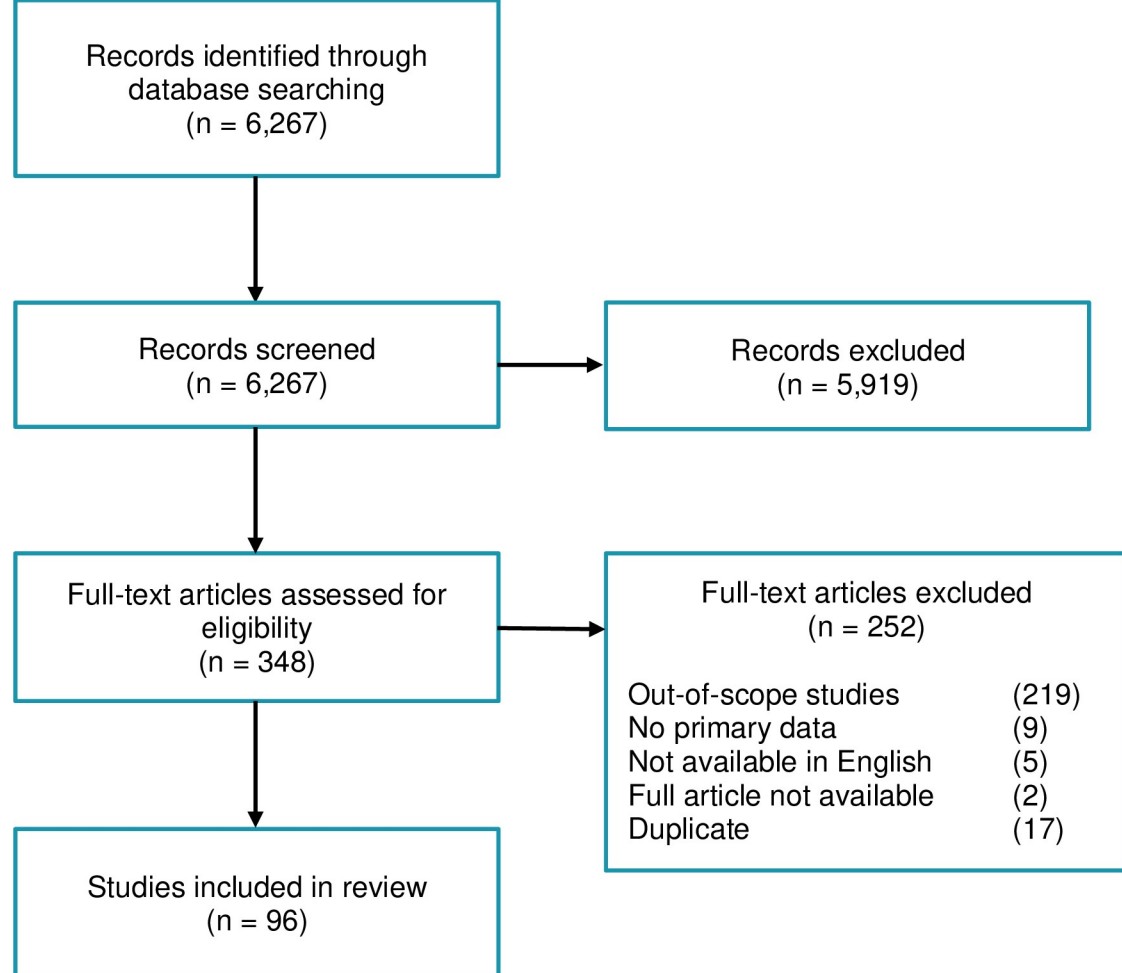

**Fig 1. Summary of screening and exclusion steps of this scoping review's methodology, and resulting number of publications after each step.**

*aegypti* were more predisposed to chikungunya infection at lower temperatures. Alto et al. [64] found that larger fluctuations in diurnal temperature range led to higher rates of chikungunya infection, and Xiao et al. [47] found that maximum dengue infection rates occurred at 31°C. Mordecai et al. [73] modelled *Ae. aegypti* and *Ae. albopictus* transmission in the Americas and found that mean temperature data accurately reflected Zika, chikungunya and dengue human case data. Transmission was found to occur between 18 and 34°C and maximal transmission was observed between 26–29°C, with less certainty surrounding the critical thermal minimum than the critical thermal maximum [73]. *Ae. albopictus* was found to perform better in cooler temperatures [73]. Buckner et al. [45] found that the interaction of low temperature and low food availability increased *Ae. aegypti* and *Ae. albopictus* susceptibility to DENV-1 serotype infection.

Three studies examined the effects of larval competition on dengue vector competence [44,45,74]. Bara et al. [44] found that *Ae. albopictus* larval competition resulted in significantly longer development times, lower emergence rates, and smaller adults, but did not significantly affect the extrinsic incubation period of DENV-2 virus. Kang et al. [74] found that larval-stage crowding and nutritional limitation led to lower survival rates until pupation, lower blood

**Table 1. Number of publications included in the scoping review, for each review section, study design, and arbovirus and mosquito vector species of interest.**

| Theme | Category | Number of publications |
|---|---|---|
| Section | Vector Species Composition | 29 |
| | Virus Transmission Dynamics | 42 |
| | Both sections | 25 |
| Study design | Field | 16 |
| | Laboratory | 40 |
| | Modelling | 27 |
| | Field and Laboratory | 9 |
| | Field and Modelling | 3 |
| | Laboratory and Modelling | 1 |
| Virus of interest | Zika | 30 |
| | Dengue | 45 |
| | Chikungunya | 10 |
| | Multiple | 11 |
| Mosquito species of interest | *Ae. aegypti* | 52 |
| | *Ae. albopictus* | 6 |
| | *Cx. quinquefasciatus* | 3 |
| | *Ae. aegypti* and *Ae. albopictus* | 19 |
| | *Ae. aegypti* and *Cx. quinquefasciatus* | 1 |
| | *Ae. albopictus* and *Cx. quinquefasciatus* | 0 |
| | *Ae. aegypti*, *Ae. albopictus* and *Cx. quinquefasciatus* | 1 |
| | Others | 12 |
| | None specifically | 2 |

feeding success, slower development, smaller adult body size, and lower susceptibility to DENV-2 infection. Four studies examined a variety of blood meal characteristics on arboviral infection rate [23,24,49,75]. Fresh Zika-infected blood meal was associated with significantly higher infection rates than frozen Zika-infected blood meal [23]. Similarly, Zika-infected whole blood meal was associated with significantly higher infection rates than Zika-infected protein meal [24]. Hill et al. [49] studied the impact of antibiotics on dengue infection rate and mosquito fertility, and found no significant association in *Ae. aegypti*. Mosquitoes exposed to DENV-2 were more likely to re-feed than those that were unexposed [75]. Sylvestre et al. [76] studied the impact of DENV-2 infection on *Ae. aegypti* life history traits, and found that it significantly affected feeding behaviour, survival, fecundity, and oviposition success.

**Vector infection rate.**    Two studies conducted in Brazil exclusively examined infection rates by Zika virus in wild mosquito populations (Table 2). Ferreira-de-Brito et al. [31] reported three Zika-infected pools of *Ae. aegypti*, but no Zika-infected *Cx. quinquefasciatus* or *Ae. albopictus* pool [31], out of 468 tested pools among the three species. Ayllón et al. [32] tested 406 *Ae. aegypti* and 11 *Ae. albopictus* field-collected individuals, and found three Zika-infected *Ae. aegypti* individuals.

Six studies reported exclusively on dengue infection rates in wild mosquito populations (Table 2). Cecílio et al. [77] observed four positive pools, out of 54 tested, among *Aedes* mosquitoes collected in two regions of Brazil over the course of 17 months, through the installation of ovitraps in public schools. Cruz et al. [57] detected eight positive pools, out of 50 *Ae. aegypti* pools, collected in Mato Grosso, Brazil. Martínez et al. [62] reported two positive pools, out of 226 *Ae. aegypti* pools, collected in Mexico. Claderón-Arguedas et al. [78] reported nine positive pools, out of 35 *Ae. albopictus* pools, collected in Costa Rica. Pérez-Pérez et al. [54] reported

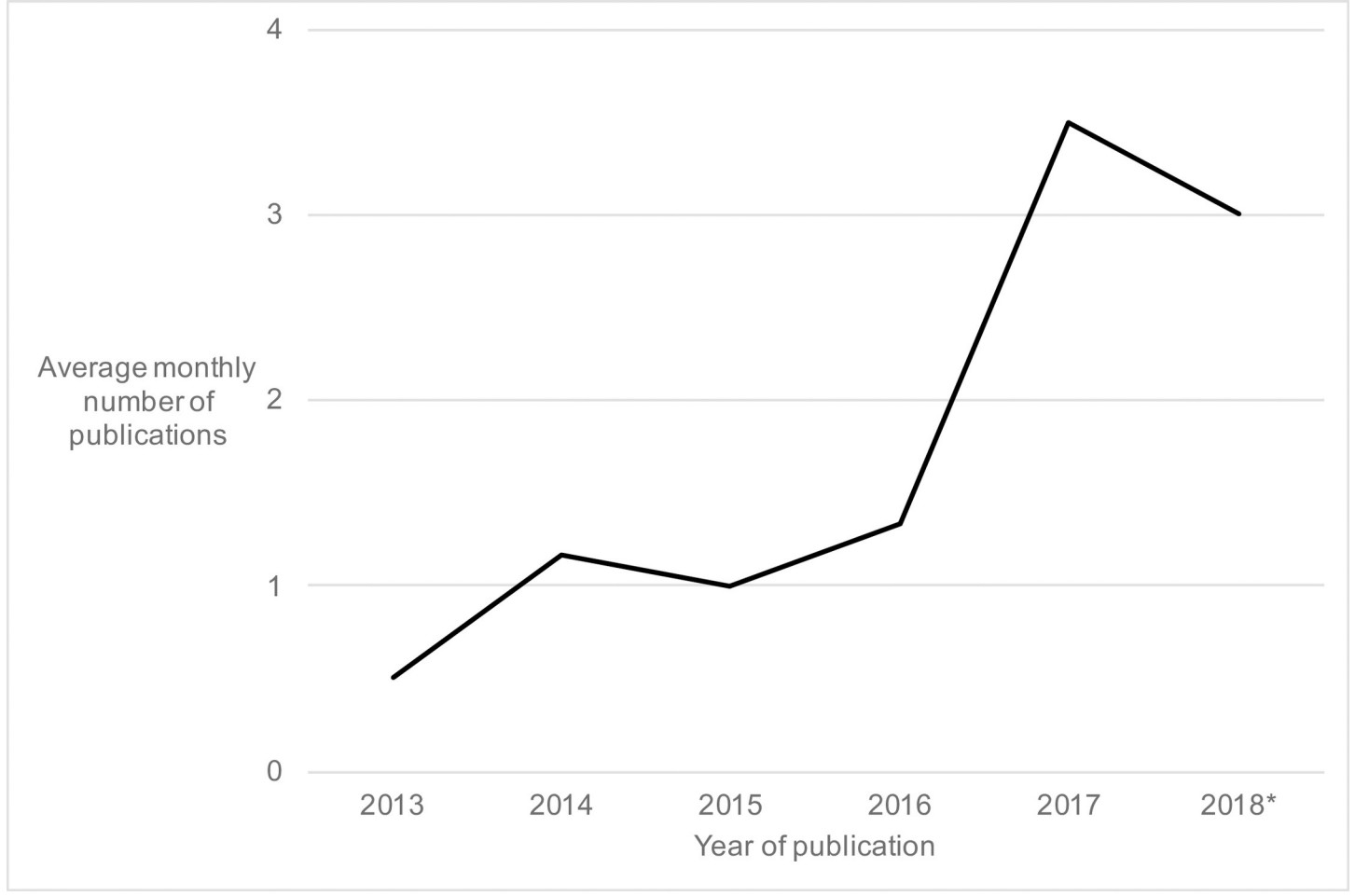

**Fig 2. Average monthly number of publications included in the scoping review, for each year since 2013, out of a total of 96.** *Year-to-date on March 1st 2018.

**Table 2. List of studies that report a proportion of positive mosquito pools for any or a combination of Zika, dengue and chikungunya viruses, along with information on authors, year and country of location of the study, and mosquito species of interest.**

| Authors | Year | Location | Mosquito species | Pools tested | Zika rate (%) | Dengue rate (%) | Chikungunya rate (%) |
|---|---|---|---|---|---|---|---|
| Ferreira-de-Brito et al. | 2016 | Brazil | *Aedes* sp. and *Cx. quinquefasciatus* | 468 | 0.64 | ø | ø |
| Ayllón et al. | 2017 | Brazil | *Ae. aegypti* and *Ae. albopictus* | 178 | 1.12 | ø | ø |
| Martínez et al. | 2014 | Mexico | *Ae. aegypti* | 226 | ø | 0.88 | ø |
| Calderón-Arguedas et al. | 2015 | Costa Rica | *Ae. albopictus* | 35 | ø | 25.71 | ø |
| Cecílio et al. | 2015 | Brazil | *Aedes* sp. | 54 | ø | 7.41 | ø |
| Cruz et al. | 2015 | Brazil | *Ae. aegypti* | 50 | ø | 16.00 | ø |
| Pérez-Castro et al. | 2016 | Colombia | *Ae. aegypti* | 34 | ø | 61.76 | ø |
| Pérez-Pérez et al. | 2017 | Colombia | *Ae. aegypti* and *Ae. albopictus* | 407 | ø | 32.43 | ø |
| Díaz-González et al. | 2015 | Mexico | *Ae. aegypti* | 557 | ø | ø | 3.23 |
| Cevallos et al. | 2018 | Ecuador | *Ae. aegypti* | 22 | 14.29 | ø | 12.50 |
| Dzul-Manzanilla et al. | 2015 | Mexico | *Ae. aegypti* | 284 | ø | 9.51 | 3.17 |
| Cigarroa-Toledo et al. | 2016 | Mexico | *Ae. aegypti* | 27–237* | ø | 0.00 | 0.84–7.40* |
| Farraudière et al. | 2017 | Martinique | *Ae. aegypti* | 414 | ø | 1.21 | 2.66 |

*Total number of pools tested is not stated, but number of sampled mosquitoes, and maximum number of mosquitoes per pool, are stated.

132 positive pools, out of 407 tested, collected in Colombia. One of the positive pools was *Ae. albopictus*, and the remainder were *Ae. aegypti*. Pérez-Castro et al. [79] reported 21 positive pools, out of 34 tested, in *Ae. aegypti* in Colombia.

A study measured the naturally-occurring prevalence of chikungunya virus in wild mosquito populations (Table 2). Díaz-González et al. [72] reported 18 *Ae. aegypti* positive pools in Mexico, out of 557 tested. A study reported on the prevalence of both chikungunya and Zika viruses among *Ae. aegypti* in Ecuador (Table 2) [30]. Three studies tested both chikungunya and dengue viruses in wild mosquito populations (Table 2). Chikungunya, but not dengue, was detected in *Ae. aegypti* in Mexico by Cigarroa-Toledo et al. [71], although both chikungunya and dengue viruses were isolated in Mexico in *Ae. aegypti* by Dzul-Manzanilla et al. [59], and in Martinique by Farraudière et al. [61].

**Vertical transmission.** Three studies reported on vertical transmission of dengue virus [58,60,80], and one [81] reported on the vertical transmission of Zika virus. Buckner et al. [80] found a vertical transmission rate of DENV-1 of 11.11% in *Ae. albopictus* and of 8.33% in *Ae. aegypti*. Da Costa et al. [58] observed dengue infection rates among third and fourth instar *Ae. aegypti* between 1.14% and 2.41% in Brazilian municipalities, and Espinosa et al. [60] observed one DENV-3 positive male *Ae. aegypti* pool, collected in Argentina. Thangamani et al. [81] experimentally injected mosquitoes with Zika virus and observed Zika virus infection in *Ae. aegypti* offspring, but not *Ae. albopictus*. Six filial *Ae. aegypti* pools out of 69 tested were found positive for Zika virus [81].

**Transmission risk modelling.** Seven studies modelled transmission dynamics for Zika virus [40,82–87]. Lourenço et al. [40] used vectorial capacity as a means of prediction, Marini et al. [82] and Majumder et al. [83] used vector abundance and human case data, and Villela et al. [84] and Ospina et al. [85] used disease notification and natural history. Rojas et al. [86] found attack rates in Girardot and San Andres, Colombia to be highest among females, aged 20–49. Fitzgibbon et al. [87] report that early host and vector heterogeneity significantly affect final epidemic size.

Eleven studies modelled dengue transmission dynamics [88–99]. Lee et al. [95] constructed a predictive model that accurately foresaw 75% of dengue outbreaks in Colombia. Reiner et al. [88] reported that social proximity drives fine-scale heterogeneity in dengue transmission rates based on data from Peru. Three studies reported that meteorological variables including temperature and humidity are important determinants of transmission dynamics [89,90, 92,93], and one study found that transovarial transmission plays an important role in transmission dynamics depending on basic reproductive number [91]. Liu-Helmersson et al. [96] predicted an increase in diurnal temperature range and increased dengue epidemic potential under climate changes in cold, temperate and extremely hot climates where mean temperatures are far from 29˚C. Velasques-Castro et al. [97] studied *Ae. aegypti* dynamics in relation to host spatial heterogeneity and generated a dengue infection risk map, based on host dynamics. Taber et al. [98] modelled the colonization of Pennsylvania by *Ae. albopictus* together with corresponding risk of dengue.

One study estimated chikungunya transmission risk according to temperature threshold for breeding and adult mosquitoes in Argentina [99]. The authors suggest that temperatures conducive to *Ae. aegypti* breeding and transmission are present during September and April in northeastern Argentina, and in January in southern Argentina. A study compared endemic and transient chikungunya and dengue transmission dynamics, and the role of virus evolution [100]. They found that reducing biting rate and vector-to-susceptible-host ratio were the most effective at reducing basic reproductive number. A study modelled transmission risk of Zika, dengue and chikungunya and found temperature data to match well with human case data [73].

**Strain infectivity and co-infection.** Six studies examined the infectivity of different dengue viral strains, and the impact of co-infection [50,74,101–104]. Muturi et al. [50] found that infection with DENV-4 rendered *Ae. aegypti* significantly less susceptible to secondary infection with DENV-2. Kang et al. [74] modelled interactions between dengue viral serotypes. Quiner et al. [101] studied the infectivity of different isolates of DENV-2, and found NI-2B to have a replicative advantage over NI-1 until 12 days following infection, after which the advantage had dissipated. Quintero-Gil et al. [102] found that the DENV-2 serotype performed with a thousand-fold greater efficiency than the DENV-3 serotype, upon co-infection. In parallel, Serrato-Salas et al. [103] found that *Ae. aegypti* were significantly less susceptible to secondary dengue infection, after having been challenged with an inactive version of the virus. Vazeille et al. [104] found that DENV-4 outperformed DENV-1 in *Ae. aegypti* upon co-infection. Nuckols et al. [46] artificially infected *Ae. aegypti* and *Ae. albopictus* with chikungunya and DENV-2 simultaneously, separately, and in reverse order. Simultaneous dissemination was detected in all groups upon co-infection, and co-transmission occurred at low rates [46]. Rückert et al. [27] found that the co-infection of *Ae. aegypti* with Zika, chikungunya and dengue viruses minimally affected vector competence, and that vectors were able to transmit each viral pair, as well as three viruses simultaneously. Alto et al. [69] found *Ae. aegypti* and *Ae. albopictus* to be susceptible to Indian Ocean and Asian chikungunya virus genotypes.

**Human disease risk.** Five articles studied correlations between entomological measures and risk of human dengue infection [105–109]. One study conducted in Peru found that *Ae. aegypti* density was not associated with an increased risk of seroconversion [105]. One study in Acre, Brazil found that *Ae. aegypti* density and risk of dengue increased with tourism and case importation [106]. A study in Mexico City found a positive correlation between dengue incidence and *Ae. aegypti* indoor abundance, as well as monthly average temperature and rainfall [107]. Another study conducted in Peru found that an individual's likelihood of being bitten in the home was directly proportional to time spent in the home, and body surface area. They did not find age or gender to be significant predictors [108]. Oliveira et al. [109] reported the circulation of four dengue serotypes in Brazil introduced between 2001 and 2012 (DENV-1, DENV-2, DENV-3, DENV-4) and reported an increase in dengue infection in Brazil during that time period, i.e. 587 cases/100 000 in 2001 to 1561 cases/100 000 in 2012. Monaghan et al. [110] predicted the seasonal abundance of *Ae. aegypti* in the United States using meterorologically driven models as a means of estimating arboviral infection risk [110]. All 50 included cities were found to be suitable during the summer months (July to September), while only cities in Florida and Texas were found to have *Ae. aegypti* abundance potential during the winter months (December to March). Lo and Park [111] found that regions of Brazil with elevated temperature and precipitation were more conducive to *Ae. aegypti* presence and Zika virus cases. Da Cruz Ferreira et al. [112] found that dengue occurrence increased by 25% when the average number of mosquitoes caught by traps increased by 0.1 per week. Stewart-Ibarra and Lowe [113] assessed the effect of climatic and entomological variables on intra-annual variability in dengue incidence in Southern Ecuador. Da Rocha Taranto et al. [114] examined the relationship between vector collection, species composition, hatching rates, and population density on dengue incidence. Hatching rate was found to be affected by population density and climate, and presence of vectors was associated with dengue incidence [114]. Ernst et al. [94] found no correlation between *Ae. aegypti* density and human age structure between two cities with different dengue transmission dynamics.

## Discussion

Our scoping review included studies focused on vector species composition and arbovirus transmission dynamics of Zika, dengue and/or chikungunya in the Americas. We observed a steady

increase in number of publications, from 2013 to 2018, with half of all studies published from January 2017 to March 2018. Sightly less than half of all studies included in this review were specifically pertaining to virus transmission dynamics. Around a third of all studies addressed vector species composition. The remainder treated aspects of both sections. Most studies focused on *Aedes aegypti* as the vector species of interest, had an exclusively laboratory-based or modelling-based study framework, and focused exclusively on either Zika or dengue. One limitation of our study is the use of a single search engine, PubMed, which may have reduced the number of included publications in our scoping review. However, given the focus of our scoping review, we believe this search engine should have captured almost all, if not all, relevant studies.

To determine vector competence, a species must be able to acquire, maintain, and transmit a pathogen, which is assessed through experimental infection studies. However, these studies are heterogeneous in both the mosquito populations and virus strains used, as well as methods measuring potential to transmit [115]. The detection of viral particles in wild-caught mosquitoes does not signify vector competence on its own, but it lends support to evidence from laboratory studies, when coupled with the observation of human host-feeding behaviour. Field studies are also important to assess the relative importance of competent vector species in disease maintenance and/or transmission. Vector competence for Zika virus has been well established for *Ae. aegypti* [19–32] and *Ae. albopictus* [22,23], but there is a growing consensus that *Cx. quinquefasciatus* is not a competent Zika virus vector, and no consensus has been reached regarding the competence of *Ae vexans*. A number of studies report that *Cx. quinquefasciatus* is refractory to Zika virus [29,36–39,41,116]. While Zika virus has been detected in a small number of field-caught *Cx. quinquefasciatus* in Brazil [42], this does not necessarily indicate their ability to transmit the virus. Interestingly, information on Zika virus vector species composition was abundant, but sparse on Zika virus transmission dynamics. Few studies examined natural infection rates of Zika virus [31,32], vertical transmission [81], or co-infection with other viruses [27]. This is in contrast to the wealth of research available on natural infection and co-infection for dengue and chikungunya viruses, although vertical transmission research was sparse for all three viruses [46,50,58,77,80,101,102].

Based on the internationally recognized urgency of Zika virus infection as a public health concern, and potential increase in the importance of this and other emerging arboviruses in the future, further research on Zika virus transmission dynamics is of pressing need. Also, given the ongoing co-circulation of these three globally spreading arboviruses in the Americas, and the resulting complexity of their transmission dynamics, more integrative studies are needed that investigate a combination of Zika, dengue and chikungunya viruses and use a variety of approaches to answer questions relating to the risk posed by these arboviruses.

## Supporting information

**S1 Table. List of full-text articles included in the review.** Information on first author's last name, year of publication, title, journal, review section, study design, and arbovirus and mosquito vector species of interest are given for each full-text article.
(XLSX)

**S2 Table. PRISMA-ScR checklist.** Checklist stating location of each element of the scoping review, as implemented by Tricco et al. [18].
(PDF)

## Acknowledgments

This research is part of an international project entitled 'Research on Arbovirus Dynamics and Mitigation–Latin America and Canada' (RADAM-LAC), with field study sites in Colombia,

Ecuador and Argentina. The RADAM-LAC Research Team consists of Beate Sander, Camila González, Jianhong Wu, Manisha A. Kulkarni, Marcos Miretti, Mauricio Espinel and Varsovia Cevallos.

## Author Contributions

**Conceptualization:** Reilly Jones, Manisha A. Kulkarni, Benoit Talbot.

**Data curation:** Reilly Jones, Thomas M. V. Davidson, Benoit Talbot.

**Formal analysis:** Reilly Jones, Thomas M. V. Davidson.

**Funding acquisition:** Manisha A. Kulkarni, RADAM-LAC Research Team.

**Investigation:** Manisha A. Kulkarni, RADAM-LAC Research Team.

**Methodology:** Reilly Jones, Manisha A. Kulkarni, Benoit Talbot.

**Project administration:** Manisha A. Kulkarni, RADAM-LAC Research Team.

**Resources:** Manisha A. Kulkarni.

**Supervision:** Benoit Talbot.

**Writing – original draft:** Reilly Jones, Thomas M. V. Davidson.

**Writing – review & editing:** Manisha A. Kulkarni, RADAM-LAC Research Team, Benoit Talbot.

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
