## [Decision Letter · Decision Letter 0]

4 Oct 2019

PONE-D-19-19188

Arbovirus Vectors of Epidemiological Concern in the Americas: A Scoping Review of Entomological Studies on Zika, Dengue and Chikungunya Virus Vectors

PLOS ONE

Dear Dr. Talbot,

Thank you very much for submitting your manuscript "Arbovirus Vectors of Epidemiological Concern in the Americas: A Scoping Review of Entomological Studies on Zika, Dengue and Chikungunya Virus Vectors" (#PONE-D-19-19188) for review by PLOS ONE. As with all papers submitted to the journal, your manuscript was fully evaluated by academic editor (myself) and by independent peer reviewers. The reviewers appreciated the attention to an important health topic, but they raised substantial concerns about the paper that must be addressed before this manuscript can be accurately assessed for meeting the PLOS ONE criteria. Therefore, if you feel these issues can be adequately addressed, we invite you to submit a revised version of the manuscript that addresses the points raised during the review process. We can’t, of course, promise publication at that time.

We would appreciate receiving your revised manuscript by Nov 18 2019 11:59PM. To enhance the reproducibility of your results, we recommend that if applicable you deposit your laboratory protocols in protocols.io, where a protocol can be assigned its own identifier (DOI) such that it can be cited independently in the future. For instructions see: http://journals.plos.org/plosone/s/submission-guidelines#loc-laboratory-protocols

We look forward to receiving your revised manuscript.

Kind regards,

Abdallah M. Samy, PhD

Academic Editor

PLOS ONE

**Journal Requirements:**

2. We ask that you provide further reporting in your methods section:

*Please provide a rationale for the dates included in your search, and why papers before 2013 were not included.

*Please explain in more detail the data that were extracted and how this was performed.

*Please explain the reasoning behind using only one electronic database for your search.

Thank you for your attention to these requests.

**Additional Editor Comments:**

I invited and received two reviews of your manuscript; both reviews raised some substantial concerns for your manuscript as it currently stands. I completely agree to the point raised by reviewers that your study missed many literature including ones that we published later (#check Samy *et al*. 2016). Please respond properly for all comments raised by our reviewers below. I would kindly ask you to check the Journal style requirements before submitting a revised version of your manuscript.

**Reviewers' comments:**

Reviewer's Responses to Questions

**Comments to the Author**

1. Is the manuscript technically sound, and do the data support the conclusions?

Reviewer #1: Yes

Reviewer #2: Yes

2. Has the statistical analysis been performed appropriately and rigorously? 

Reviewer #1: N/A

Reviewer #2: N/A

3. Have the authors made all data underlying the findings in their manuscript fully available?

Reviewer #1: Yes

Reviewer #2: Yes

4. Is the manuscript presented in an intelligible fashion and written in standard English?

Reviewer #1: Yes

Reviewer #2: Yes

5. Review Comments to the Author

Reviewer #1: The manuscript is a scoping review that provides an overview on the current state of entomological knowledge of the mosquito vectors of Zika, dengue and chikungunya viruses in the Americas. The authors performed a thorough review of articles on these viruses and theirs hosts published between January 2013 and March 2018. I believe that this study represents a valuable overview of a wide body of research. However, I believe that the study does not entirely meet its own objectives, and could benefit from a more focused approach and a bit more synthesis. I am also concerned that the short time-frame examined by the authors does not provide a full and accurate picture of the literature on several topics. While the nature of publishing makes it impossible for a review to capture all of the most up-to-date papers, the time frame set by the authors starts too late to capture many valuable studies on important vector species. The authors’ time frame seems suited to a study of public health effects of these arboviruses in the Americas, but there is much more entomological literature from the ‘90s and 2000s that would provide a much more complete picture on this subject. I would strongly encourage the authors to broaden their time frame.

A few comments and suggestions are listed below:

Abstract:

1) Line 28: Please correct the spelling of “arthropod”

2) Line 34: I don’t know that I agree with this. While Ae. albopictus has demonstrated vector competence for all three viruses, and has been implicated in the transmission of these viruses in Europe, Asia and the Americas, there is still little evidence that it is the primary vector for any of these viruses in the Americas. Please remove the word “primary”.

3) Line 49: This seems like a bit of a leap, given that the authors list several good studies on interspecific competition. Perhaps it would be better to say simply that much work is left to be done regarding interspecific competition, given the breadth of the topic.

4) General comments: I would like to see some mention of methods in the abstract. Also, it seems redundant to have both “Key findings” and “Conclusions,” especially given how short the “Conclusions” section is.

Introduction:

5) Lines 61-63: This language is somewhat confusing. It implies that ZIKV, DENV and CHIKV are the only arboviruses with global health implications. It also makes it somewhat unclear that "arboviruses" and "arthropod-borne viruses" are the same thing.

6) Line 68: What is the meaning of “uncharacterized” here? Perhaps it would be better to say that until recently ZIKV transmission has been confined to Africa and Asia.

7) Line 93: Arthralgia can actually persist for over a year, if not longer. See Gianandrea et al., 2008.

8) Line 109: I’m not sure about the term “evidence” here. Perhaps “knowledge” would be more accurate?

Methods:

9) Lines 117-118: I would suggest that the authors consider using an additional search library, such as Web of Science. It might provide a broader pool of results.

10) General comment: Was this study conducted using PRISMA guidelines? If so, please mention this and provide PRISMA checklist.

Results:

11) Lines 200-205: This section is a prime example of why this paper could benefit from broadening the examined time frame. There are papers from Richards et al. (2006), Dennett et al. (2007), and Niebylski et al. (1994) that provide a much more complete picture of Ae. albopictus blood feeding behavior in North America.

12) Lines 293-299: Medley et al. (2014) also provides valuable insight into Ae. albopictus population genetics across North America. Not sure why this did not meet study criteria.

Discussion:

13) Line 103: Please replace “immatures” with a more correct term. “immature stages” or “larvae” would be fine.

14) Line 108: “have been consistently with arbovirus vector occurrence”… Consistently associated? Please clarify.

15) Line 111: Again, I do not think it is fair to say that it is poorly understood, though more research is certainly needed. Please rephrase.

General overall comments:

16) Please review manuscript for spelling errors and punctuation.

17) While the manuscript lays out much of the research around ZIKV, DENV, CHIKV and vector species, it does not provide much in the way of highlighting knowledge gaps. I think this would be a much stronger paper if there were more emphasis in the discussion on specifically highlighting issues that need more research to resolve.

Reviewer #2: In this manuscript the authors provide a review of entomological studies of three emerging arboviruses of critical concern in the past decade. Though this sort of review has been conducted before, this manuscript distinguishes itself by providing a framework for identifying knowledge gaps and guiding future research.

This scoping review is certainly informative, but it reads more as a perspective piece than a research article. While there is nothing wrong with that, it should be clearer in its presentation of that fact.

Even in a scoping review article, I believe a manuscript should still address key questions and I felt that this review read more as a book chapter than a research article. While the authors do include the question “What is the current state of the evidence on mosquito vector population dynamics and behaviour, mosquito vector distribution and environmental suitability, vector species composition and virus transmission by mosquito vectors as they relate to Zika, dengue

and chikungunya viruses in the Americas?” this is a very broad overarching inquiry. I would have preferred specific directed questions which are supported by the scoping review. For example, Are Ae. aegypti and Ae. albopictus distributions influenced by specific environmental conditions? Is this uniform globally? As a scoping review which aims to highlight knowledge gaps I felt that this kind of structure would be beneficial.

I would have liked to see more directed questions.

Abstract

-arthropod is misspelled

-Why were only studies from 2013 considered? Because of the shift in focus to research on these viruses? What about studies between 2013 and 2016 which were covered in Waddell et al.

-How much overlap is there between the 2016 scoping review and this one?

LN 62: yes largely, but new findings highlight the importance of ticks and other arthropods, otherwise we could call them mosquito-borne viruses. Please modify to specify that many are transmitted by mosquitoes instead of largely.

LN 93: background information on chikungunya seems pale in comparison to that of Zika and dengue. Since this is not just a scoping review of Zika, I would expand the section on chikungunya significantly

Ln 95: sub-Saharan Africa

LN 101: please include

Kraemer, M.U., Reiner, R.C., Brady, O.J., Messina, J.P., Gilbert, M., Pigott, D.M., Yi, D., Johnson, K., Earl, L., Marczak, L.B. and Shirude, S., 2019. Past and future spread of the arbovirus vectors Aedes aegypti and Aedes albopictus. Nature microbiology, 4(5), p.854.

While the statement in LN 205: “Vinauger et al. [29] experimentally demonstrated the importance of dopamine in relation to host-seeking behaviour” is interesting, this falls more within behavior than blood feeding. Host seeking and feeding are different strategies and dopamine being an attractant may not influence patterns of blood feeding unless you have species differences in the release of dopamine.

LN 154: chikungunya should be lower case

LN 200: No comment is made about Culex

The subsections of the results are intriguing, but I find that some sections are lacking in content beyond one or two studies.

LN 334: there have been conflicting studies on socioeconomics and Aedes vectors and geography

This review provides a long list of directions in which future researchers can begin to investigate and this review is well outlines and compiled; however, all of the outstanding questions seem scattered throughout the text. I would prefer to see a conceptual figure outlining the different disciplines and specific questions remaining. If not a figure, a table would be a very useful guide or framework for researchers to build off of.

6. PLOS authors have the option to publish the peer review history of their article (what does this mean?). If published, this will include your full peer review and any attached files.

Reviewer #1: No

Reviewer #2: No

---

## [Author Response · Author response to Decision Letter 0]

15 Nov 2019

Here you can find the list of comments from the editor and two reviewers, in black font, with our response indented below each. The respective changes are highlighted in the edited manuscript file.

Additional Editor Comments:

I invited and received two reviews of your manuscript; both reviews raised some substantial concerns for your manuscript as it currently stands. I completely agree to the point raised by reviewers that your study missed many literature including ones that we published later (#check Samy et al. 2016). Please respond properly for all comments raised by our reviewers below. I would kindly ask you to check the Journal style requirements before submitting a revised version of your manuscript.

 We thank the editor for their suggestion of an additional publication that seems to be missing from our reference list. However, it is important to point out that our inclusion criteria specified studies that analyzed some primary data. It seems the suggested reference solely analyzes secondary data. According to comments of the two reviewers, we decided to adopt a more focused approach to frame our scoping review. We modified a sentence at lines 8-10 of page 6 to reflect this.

Reviewers' comments:

Reviewer #1: The manuscript is a scoping review that provides an overview on the current state of entomological knowledge of the mosquito vectors of Zika, dengue and chikungunya viruses in the Americas. The authors performed a thorough review of articles on these viruses and theirs hosts published between January 2013 and March 2018. I believe that this study represents a valuable overview of a wide body of research. However, I believe that the study does not entirely meet its own objectives, and could benefit from a more focused approach and a bit more synthesis. I am also concerned that the short time-frame examined by the authors does not provide a full and accurate picture of the literature on several topics. While the nature of publishing makes it impossible for a review to capture all of the most up-to-date papers, the time frame set by the authors starts too late to capture many valuable studies on important vector species. The authors’ time frame seems suited to a study of public health effects of these arboviruses in the Americas, but there is much more entomological literature from the ‘90s and 2000s that would provide a much more complete picture on this subject. I would strongly encourage the authors to broaden their time frame.

 We thank the reviewer for their useful suggestions. We agree that the timeframe of the review does not capture many earlier studies in the entomological literature pertaining to arbovirus vectors, and have therefore reframed our scoping review’s questions to focus more specifically on identifying and characterizing the literature related to vector species composition and arbovirus transmission dynamics in a region of recent arbovirus introduction and ongoing co-circulation. This justification for including studies from 2013 is now stated in the Introduction on lines 5-9 of page 5. 

A few comments and suggestions are listed below:

Abstract:

1) Line 28: Please correct the spelling of “arthropod”

 We fixed the word in question.

2) Line 34: I don’t know that I agree with this. While Ae. albopictus has demonstrated vector competence for all three viruses, and has been implicated in the transmission of these viruses in Europe, Asia and the Americas, there is still little evidence that it is the primary vector for any of these viruses in the Americas. Please remove the word “primary”.

 We fixed the sentence in question, and removed the word “primary” from it.

3) Line 49: This seems like a bit of a leap, given that the authors list several good studies on interspecific competition. Perhaps it would be better to say simply that much work is left to be done regarding interspecific competition, given the breadth of the topic.

 We fixed the sentence in question.

4) General comments: I would like to see some mention of methods in the abstract. Also, it seems redundant to have both “Key findings” and “Conclusions,” especially given how short the “Conclusions” section is.

Introduction:

 We merged the abstract sections “Key findings” and “Conclusions” into a section named “Key findings”. We also added a “Methods” abstract section.

5) Lines 61-63: This language is somewhat confusing. It implies that ZIKV, DENV and CHIKV are the only arboviruses with global health implications. It also makes it somewhat unclear that "arboviruses" and "arthropod-borne viruses" are the same thing.

 We fixed the sentence in question.

6) Line 68: What is the meaning of “uncharacterized” here? Perhaps it would be better to say that until recently ZIKV transmission has been confined to Africa and Asia.

 We fixed the sentence in question.

7) Line 93: Arthralgia can actually persist for over a year, if not longer. See Gianandrea et al., 2008.

 We fixed the sentence in question.

8) Line 109: I’m not sure about the term “evidence” here. Perhaps “knowledge” would be more accurate?

 We fixed the sentence in question.

Methods:

9) Lines 117-118: I would suggest that the authors consider using an additional search library, such as Web of Science. It might provide a broader pool of results.

 We thank the reviewer for their suggestion. We have reframed the review to focus more closely on vector species composition and arbovirus transmission dynamics with implications for public health, and as such we feel that the Pubmed search engine is suitable to identify relevant primary literature for our scoping review. Our search terms and inclusion criteria were intentionally broad to capture a wide range of studies. However, we have mentioned this as a limitation in the discussion, at lines 5-9 of page 17.

10) General comment: Was this study conducted using PRISMA guidelines? If so, please mention this and provide PRISMA checklist.

 We added a sentence to state that we used PRISMA guidelines, at lines 14-15 of page 5.

Results:

11) Lines 200-205: This section is a prime example of why this paper could benefit from broadening the examined time frame. There are papers from Richards et al. (2006), Dennett et al. (2007), and Niebylski et al. (1994) that provide a much more complete picture of Ae. albopictus blood feeding behavior in North America.

 According to a previous comment by the reviewer, we adopted a more focused approach for our scoping review. With this new focus, we decided to remove the review section in question. 

12) Lines 293-299: Medley et al. (2014) also provides valuable insight into Ae. albopictus population genetics across North America. Not sure why this did not meet study criteria.

 According to a previous comment by the reviewer, we adopted a more focused approach for our scoping review. With this new focus, we decided to remove the review section in question.

Discussion:

13) Line 103: Please replace “immatures” with a more correct term. “immature stages” or “larvae” would be fine.

 We fixed the sentence in question.

14) Line 108: “have been consistently with arbovirus vector occurrence”… Consistently associated? Please clarify.

 We fixed the sentence in question.

15) Line 111: Again, I do not think it is fair to say that it is poorly understood, though more research is certainly needed. Please rephrase.

 We removed the sentence in question.

General overall comments:

16) Please review manuscript for spelling errors and punctuation.

 We made a thorough spelling and punctuation check of the entire manuscript..

17) While the manuscript lays out much of the research around ZIKV, DENV, CHIKV and vector species, it does not provide much in the way of highlighting knowledge gaps. I think this would be a much stronger paper if there were more emphasis in the discussion on specifically highlighting issues that need more research to resolve.

 According to a previous comment by the reviewer, we adopted a more focused approach for our scoping review. We reframed the discussion to reflect these changes and added more focus to highlight knowledge gaps.

Reviewer #2: In this manuscript the authors provide a review of entomological studies of three emerging arboviruses of critical concern in the past decade. Though this sort of review has been conducted before, this manuscript distinguishes itself by providing a framework for identifying knowledge gaps and guiding future research.

This scoping review is certainly informative, but it reads more as a perspective piece than a research article. While there is nothing wrong with that, it should be clearer in its presentation of that fact.

Even in a scoping review article, I believe a manuscript should still address key questions and I felt that this review read more as a book chapter than a research article. While the authors do include the question “What is the current state of the evidence on mosquito vector population dynamics and behaviour, mosquito vector distribution and environmental suitability, vector species composition and virus transmission by mosquito vectors as they relate to Zika, dengue

and chikungunya viruses in the Americas?” this is a very broad overarching inquiry. I would have preferred specific directed questions which are supported by the scoping review. For example, Are Ae. aegypti and Ae. albopictus distributions influenced by specific environmental conditions? Is this uniform globally? As a scoping review which aims to highlight knowledge gaps I felt that this kind of structure would be beneficial.

I would have liked to see more directed questions.

 We thank the reviewer for their suggestion. According to a comment by another reviewer and also this comment by the reviewer, we have reframed our scoping review’s questions to focus more specifically on identifying and characterizing the literature related to vector species composition and arbovirus transmission dynamics in a region of recent arbovirus introduction and ongoing co-circulation. This justification for including studies from 2013 is now stated in the Introduction on lines 5-9 of page 5.

Abstract

-arthropod is misspelled

 We fixed the word in question.

-Why were only studies from 2013 considered? Because of the shift in focus to research on these viruses? What about studies between 2013 and 2016 which were covered in Waddell et al.

-How much overlap is there between the 2016 scoping review and this one?

 According to a comment by another reviewer and this comment by the reviewer, we have reframed our scoping review’s questions to focus more specifically on identifying and characterizing the literature related to vector species composition and arbovirus transmission dynamics in a region of recent arbovirus introduction and ongoing co-circulation. This justification for including studies from 2013 is now stated in the Introduction on lines 5-9 of page 5. 

 Also, it is important to point out that no study about vector species composition and virus transmission dynamics included in Waddell et al.’s paper was specifically related to ongoing circulation in the Americas, which was one of our selection criteria. Therefore, there is no overlap of studies between our scoping review and Waddell et al.’s. We modified a sentence at lines 1-5 of page 5 to state this.

LN 62: yes largely, but new findings highlight the importance of ticks and other arthropods, otherwise we could call them mosquito-borne viruses. Please modify to specify that many are transmitted by mosquitoes instead of largely.

 We fixed the sentence in question.

LN 93: background information on chikungunya seems pale in comparison to that of Zika and dengue. Since this is not just a scoping review of Zika, I would expand the section on chikungunya significantly

 We acknowledge the section on Zika virus is much lengthier than the section on dengue and chikungunya viruses. We decided to remove some of the context information about Zika which is not directly important to set our scoping review in context. The extent of background information for all three arboviruses is now similar. 

Ln 95: sub-Saharan Africa

 We fixed the word in question.

LN 101: please include

Kraemer, M.U., Reiner, R.C., Brady, O.J., Messina, J.P., Gilbert, M., Pigott, D.M., Yi, D., Johnson, K., Earl, L., Marczak, L.B. and Shirude, S., 2019. Past and future spread of the arbovirus vectors Aedes aegypti and Aedes albopictus. Nature microbiology, 4(5), p.854.

 We included the specific reference in the sentence in question.

While the statement in LN 205: “Vinauger et al. [29] experimentally demonstrated the importance of dopamine in relation to host-seeking behaviour” is interesting, this falls more within behavior than blood feeding. Host seeking and feeding are different strategies and dopamine being an attractant may not influence patterns of blood feeding unless you have species differences in the release of dopamine.

 According to a previous comment by the reviewer, we adopted a more focused approach for our scoping review. With this new focus, we decided to remove the review section in question.

LN 154: chikungunya should be lower case

 We fixed the word in question.

LN 200: No comment is made about Culex

 The subsections of the results are intriguing, but I find that some sections are lacking in content beyond one or two studies.

LN 334: there have been conflicting studies on socioeconomics and Aedes vectors and geography

 According to a previous comment by the reviewer, we adopted a more focused approach for our scoping review. With this new focus, we decided to remove the review sections in question.

---

## [Editor Report · Decision Letter 1]

7 Jan 2020

Arbovirus Vectors of Epidemiological Concern in the Americas: A Scoping Review of Entomological Studies on Zika, Dengue and Chikungunya Virus Vectors

PONE-D-19-19188R1

Dear Dr. Talbot,

We are pleased to inform you that your manuscript has been judged scientifically suitable for publication and will be formally accepted for publication once it complies with all outstanding technical requirements.

With kind regards,

Abdallah M. Samy, PhD

Academic Editor

PLOS ONE

---

## [Editor Report · Acceptance letter]

23 Jan 2020

PONE-D-19-19188R1 

Arbovirus Vectors of Epidemiological Concern in the Americas: A Scoping Review of Entomological Studies on Zika, Dengue and Chikungunya Virus Vectors 

Dear Dr. Talbot:

I am pleased to inform you that your manuscript has been deemed suitable for publication in PLOS ONE. Congratulations! Your manuscript is now with our production department. 

With kind regards,

on behalf of

Dr. Abdallah M. Samy 

Academic Editor

PLOS ONE